# Digenic Congenital Hypogonadotropic Hypogonadism Due to Heterozygous *GNRH1* p.R31C and *AMHR2* p.G445_L453del Variants

**DOI:** 10.3390/genes14061204

**Published:** 2023-05-31

**Authors:** Bronwyn G. A. Stuckey, Timothy W. Jones, Bryan K. Ward, Scott G. Wilson

**Affiliations:** 1Keogh Institute for Medical Research, Nedlands, WA 6009, Australia; 2Department of Endocrinology and Diabetes, Sir Charles Gairdner Hospital, Nedlands, WA 6009, Australia; 3Medical School, University of Western Australia, Nedlands, WA 6009, Australia; 4Telethon Kids Institute, Nedlands, WA 6009, Australia; 5Perth Children’s Hospital, Nedlands, WA 6009, Australia; 6Harry Perkins Institute for Medical Research and Centre for Medical Research, University of Western Australia, Nedlands, WA 6009, Australia; 7School of Biomedical Sciences, University of Western Australia, Nedlands, WA 6009, Australia

**Keywords:** minipuberty, congenital hypogonadotropic hypogonadism, *GNRH1*, *AMHR2*, Kallmann syndrome

## Abstract

A 28-year-old man with congenital hypogonadotropic hypogonadism (CHH) was found to be heterozygous for the *GNRH1* p.R31C mutation, reported in the literature as pathogenic and dominant. The same mutation was found in his son at birth, but the testing of the infant at 64 days confirmed the hormonal changes associated with minipuberty. This led to further genetic sequencing of the patient and his son, which found a second variant, *AMHR2* p.G445_L453del, in the heterozygous form, reported as pathogenic in the patient but not in his son. This suggests a digenic cause of the patient’s CHH. Together, these mutations are postulated to contribute to CHH by the lack of anti-Müllerian hormone (AMH) signalling, leading to the impaired migration of gonadotrophin releasing hormone (GnRH) neurons, the lack of the AMH effect on GnRH secretion, and altered GnRH decapeptide with reduced binding to GnRH receptors. This led us to the conclusion that the observed *GNRH1* mutation in the heterozygous state is not certain to be dominant or, at least, exhibits incomplete penetrance and variable expressivity. This report also emphasises the opportunity afforded by the time window of minipuberty in assessing the inherited genetic disorders of hypothalamic function.

## 1. Introduction

The proper functioning of the hypothalamic–pituitary–gonadal (HPG) axis depends upon a cascade of stimulatory hormones. The hypothalamus controls pituitary gonadotrophin release by the pulsatile secretion of GnRH from GnRH neurons. In gestation during embryogenesis, GnRH neurons migrate from the olfactory placode to the preoptic-hypothalamic continuum, from whence they send projections to the median eminence to secrete GnRH and stimulate gonadotropin release by the pituitary. Failure of this process during embryological development results in congenital hypogonadotropic hypogonadism (CHH).

Multiple genes control the proper development and function of this pathway, and variants of those genes may be associated with CHH, either monogenic, digenic, or oligogenic, and in recessive, additive, or dominant inheritance patterns [1]. Over 60 genes have been implicated in the aetiology of CHH, with or without nonreproductive phenotypes. In general, the gene variants responsible for CHH may affect the HPG axis at three levels: (a) GnRH neuronal migration (*ANOS1*, *FGFR1/FGF8*, *PROKR2/PROK2*, *CHD7*, *HS6ST1*, *NMSF/NELF*, and *SEMA3A*); (b) defects in the synthesis and secretion of GnRH (*GNRH1*, *KISS1R/KISS1*, and *TACR3/TAC3*); and (c) defects in the pituitary gonadotrophs (*GNRHR*, *LHB*, and *FSHB*) [2].

Males with CHH may present at three significant time points in life. They may present at birth with features of impaired androgen effect (cryptorchidism or micropenis) [3]. The inguinoscrotal phase of testicular descent requires androgen production, and in clinical practice, the increased prevalence of cryptorchidism, with or without micropenis, in boys with CHH highlights the crucial role of androgens in this second phase of testicular descent and an opportunity to diagnose CHH early in life [4,5].

However, commonly, males with CHH may present in their teenage years, with failure to progress into puberty and no history of a neonatal phenotype to suggest CHH, and are often interpreted as having constitutional pubertal delay. In these cases, the diagnosis may be delayed substantially, even into the early twenties, compromising future adult health, psychosocial development, bone structure, and fertility.

Finally, males with CHH may present still later in life with infertility. Although the induction of spermatogenesis is possible with exogenous gonadotrophins, the undeveloped HPG axis requires a prolonged course of gonadotrophin therapy before the appearance of sperm in the ejaculate [6]. In this respect, crucially, the phenomenon of minipuberty is responsible for priming the testis for future fertility. In the weeks after birth, male infants have a surge of pituitary gonadotrophins driven by GnRH. This surge begins soon after birth and continues until 6 months of age, and is accompanied by an LH-driven rise in testosterone and by an FSH-driven four-fold increase in the number of Sertoli cells [7]. The number of Sertoli cells is critical for subsequent sperm-producing capacity because each Sertoli cell can only support the development of a finite number of germ cells. The presence or absence of minipuberty is postulated, therefore, to have a profound influence on Sertoli cell numbers and the course of the induction of spermatogenesis in later life [8]. This critical period of testicular development is missing in infants with CHH and is the likely cause of the lengthy course of fertility treatment before successful sperm production can be achieved.

This period encompassing minipuberty is an opportune time in which to measure hormones if a pituitary or hypothalamic abnormality is suspected because of clinical features in the baby associated with CHH, or, as in this case report, in the presence of a known familial mutation associated with CHH. It is a time when the diagnosis of CHH may be made early in life. It is also, as in our case, a time when one can assess the effect of a known inherited mutation on the functioning of the HPG axis. The hormonal assessment of minipuberty can also, as in our case, alert the clinician to the likelihood of a digenic cause of the parental clinical phenotype, hitherto unsuspected.

## 2. Case Report

A 28-year-old man presented for fertility treatment.

He had initially presented at age 17 with failure of progression into puberty. His height at that time was 191 cm, facial and body hair were absent, and there was no gynaecomastia. The testes were 5 mL and 6 mL, penile development stage 2, and pubic hair stage 3. There was no history of cryptorchidism. His sense of smell was intact. Pituitary MRI and other pituitary hormones were normal. His bone age was delayed at 14 years. Gonadotrophins and testosterone were low, with LH 0.8 (reference range: 1–5 mU/L), FSH 1.7 (reference range: 1–5 mU/L), and testosterone < 1.0 (reference range: 1–25 nmol/L). On LHRH stimulation, the levels of testosterone and gonadotrophins rose, with testosterone 8 nmol/L, LH 23 mU/L, and FSH 8.6 mU/L at 60 min, indicating that the pituitary–gonadal axis was intact, and the likely cause lay in the hypothalamus. A diagnosis of congenital hypogonadotropic hypogonadism (CHH) was made. He was treated with testosterone esters to induce puberty.

At the time of his subsequent presentation for fertility at age 28, he was not on testosterone therapy. His height was 202 cm and his weight was 90 kg. The penis was Tanner stage 5, pubic hair stage 4, and each testis was 12 mL and soft in consistency. Body hair was scanty, and there was no gynaecomastia. Biochemistry was as follows: LH 2 (reference range: 1–8 mU/L), FSH 3 (reference range: 1–8 mU/L), and testosterone 1.6 (reference range: 10–35 nmol/L). Prolactin, thyroid function tests, and morning cortisol were normal. He had azoospermia. Laboratory analyses were conducted as part of clinical care, and written informed consent for the investigations was provided by the subjects or their legal guardians. Genetic sequencing revealed that the patient had the p.Arg31Cys *GNRH1* variant.

The patient and his wife then received genetic counselling advice that this *GNRH1* variant was classified as pathogenic according to ACMG guidelines, had also been reported in other unrelated families, and was thought to be autosomal dominant. This advice was based on publications reporting CHH in patients with the heterozygote p.Arg31Cys *GNRH1* mutation, albeit in some with a less severe phenotype [9,10,11].

The patient underwent the induction of spermatogenesis with human chorionic gonadotrophin (hCG, Pregnyl, Organon Pharma Pty Ltd., Macquarie Park, Australia) 1500 units and the recombinant follicle-stimulating hormone (FSH, Gonal-F, Merck Healthcare Pty Ltd., Macquarie Park, Australia) 150 units, both 3 times weekly, given subcutaneously through self-injection. Testosterone response was prompt—25 nmol/L (10–35)—but spermatogenesis was exceedingly slow, with a total sperm count of 0.6 million at 18 months and 1.5 million at 36 months. The sperm count was sufficient for assisted reproduction (in vitro fertilisation (IVF)), and the patient and his wife were referred to an IVF unit. The patient’s wife conceived via IVF with intracytoplasmic sperm injection (ICSI), using the patient’s sperm, and subsequently delivered a male infant. The preimplantation genetic testing of the implanted embryo was not performed since the possible phenotypic outcome did not reach the threshold of clinical severity and disability required for local approval by the government’s reproductive technology committee [12].

The infant had no dysmorphic features and had normal external genitalia with both testes descended, each 2 ml in volume. The infant’s genetics, analysed on genomic DNA isolated from leucocytes in cord blood at birth, revealed that he had, indeed, inherited the heterozygous *GNRH1* p.R31C mutation. For that reason, we proceeded to investigate the presence or absence of minipuberty. Reassuringly, the biochemistry analysis performed via liquid chromatography–mass spectrometry at 64 days of age confirmed biochemistry parameters consistent with minipuberty, with testosterone 6.1 nmol/L (0.3–4.0), LH 1.3 mU/L (0.0–3.9), and FSH 3.0 mU/L (<3.0).

Since the previous advice was that the p.R31C mutation of *GNRH1* was thought to be dominant in producing CHH, and therefore, minipuberty is not expected to occur, we suspected another gene to be contributing to the father’s CHH. Further genetic sequencing was performed on both the patient and his son.

## 3. Methods

Initial analysis was carried out by the hospital’s routine molecular genetics laboratory. The proband’s genomic DNA was extracted from blood leucocytes, and genetic analysis was performed using massively parallel sequencing of a panel of genes associated with CHH, namely *GNRHR*, *GNRH1*, *KISS1R*, *KISS*, *TAC3*, *TACR3*, *ANOS1*, *FGFR1*, *FGF8*, *PROK2*, *PROKR2*, *CHD7*, *HS6T1*, *WDR11*, *SEMA3A*, *SOX10*, *NSMF*, *FSHB*, and *LHB*. Sequencing was performed using the TruSight One Sequencing Kit on the Illumina NextSeq-550, with at least 95% of targeted regions covered to a depth of at least 20 fold. Illumina BWA Enrichment software (v2.1.1), utilising BWA-MEM and GATK, was used for alignment and variant calling. Variants were analysed in Bench Lab NGS (Cartagenia) and classified according to ACMG Guidelines. Only high QC pathogenic or likely pathogenic variants were Sanger-confirmed and reported. Sequencing revealed a mutation in *GNRH1* (GenBank: NM_001083111.2):c.91C>T, with predicted consequences p.Arg31Cys, which had previously been reported in the Leiden Open (source) Variation Database (LOVD; Variant #0000814744).

After the birth of the male infant and confirmation of minipuberty, further genetic sequencing for the patient and his son was performed at the Endocrinology Department’s specialist molecular endocrinology laboratory. Exome capture was performed using Agilent SureSelectXT Low Input CREv2 with subsequent sequencing (150 bp paired end) on Illumina NovaSeq. Genetic data were generated and processed using conventional exome pipelines. Specifically, sequencer image analysis was performed in real time using the NovaSeq Control Software (NCS) (v1.7.5), and the Real-Time Analysis (RTA) software v3.4.4 was employed for base calling on a NovaSeq instrument computer. The primary sequence data were generated using the bcl2fastq 2.20.0.422 and the BWA (mem) (v0.7.17-r1188) was used to map the fastq reads to the reference genome (hg19) with SAMtools software (v1.8) used to sort and compress the alignments. Variant calls were carried out with HaplotypeCaller using GATK (v4.0.4.0). These analyses resulted in 92.14% of all targets having over 20× coverage, and 73.62% of all targets with over 50×, initially yielding 536,697 genetic variants, which were filtered down to 89,027 variants. ANNOVAR was used to annotate the genetic variants that were observed. Pathogenicity interpretation was accomplished using the Genetic Variant Interpretation Tool (University of Maryland), which efficiently implements the ACMG Guidelines [13,14]. The sorting and filtering of single-nucleotide variants (SNVs) and small genomic insertions and deletions (InDels) retained only those genetic variants that had a minor allele frequency (MAF) < 0.005. Only genetic variants in genes relevant to the clinical investigation were considered in the analyses. Variants of relevance were confirmed with Sanger sequencing in both father and son.

The reanalysis of the exome data, both filtered and unfiltered, manually and with Phenolyzer software [15], was performed to confirm that no other relevant pathogenic variant was overlooked in the patient. Manual curation involved sorting and filtering based on minor allele frequency and the metrics available from ANNOVAR analysis, including Polyphen, SIFTS, MutationAssessor, CADD, GERP, and SiPhy [16,17,18,19,20,21,22]. Additional analysis with Phenolyzer software was performed. Phenolyzer analysis employs a machine learning model that integrates multiple features to score and prioritise all the genes potentially relevant to the patient’s disorder, in combination with variant pathogenicity meta-predictors [23]. The combination of these two approaches provides a thorough examination of the exome sequence data. Therefore, it seems very unlikely that any other relevant pathogenic variants would be overlooked.

## 4. Results

The heterozygous *GNRH1* p.R31C variant was reconfirmed in the proband’s genomic DNA extracted from blood leucocytes and in his son’s genomic DNA extracted from epithelial cells in a buccal swab (Figure 1). An additional heterozygous 27 nucleotide non-frameshift deletion in the anti-Müllerian hormone (AMH) receptor type 2 (*AMHR2)* gene was found in the patient but not in his son (Figure 1). *AMHR2* encodes the receptor for AMH. The variant, *AMHR2* (GenBank: NM_020547):c.1330_1356del, with predicted consequence p.Gly445_Leu453del, is also reported in LOVD (Variant #0000622936) and ClinVar (VCV000008627.3) as “pathogenic”, and was not included on the original panel of genes screened by the routine laboratory. Further family studies confirmed the *AMHR2* p.G445_L453del variant in one of the patient’s parents, but neither parent had the *GNRH1* p.R31C variant (Figure 2). Genetic testing for the allelic segregation of a panel of 21 highly polymorphic variable number tandem repeat (VNTR) polymorphisms confirmed the pedigree. The resultant conclusion is that the *GNRH1* mutation in the patient was a de novo mutation not present in a parental germline but most likely had occurred in parental gametes during either oogenesis or spermatogenesis.

Therefore, on the basis of his genetics and clinical phenotype, we postulate that CHH in the patient is digenic, produced by the combination of the *GNRH1* and *AMHR2* variants. Secondly, on the basis that his son inherited the *GNRH1* p.R31C variant only and yet achieved minipuberty, we postulate that this variant is pathogenic only under certain circumstances and therefore appears to be not dominant.

## 5. Discussion

*GNRH1* mutations are among the rarer genetic causes of CHH. However, fertility in CHH can be achieved with gonadotrophin therapy. The transmission of genetic variants and potential phenotypic abnormalities to the next generation is of clinical concern and helps to plan timely clinical intervention and care. In this case, the hormonal testing of the patient’s son, in the appropriate window of time, confirmed hormonal changes consistent with minipuberty. This led us to believe that the *GNRH1* variant was not the sole gene variant responsible for the father’s clinical phenotype. Further genetic testing led us to identify a second genetic variant in the patient, which was not present in his son, and, therefore, question whether this *GNRH1* variant is dominant.

In embryogenesis, GnRH neurons migrate from the olfactory placode to the preoptic-hypothalamic continuum, from whence they send projections to the median eminence to secrete GnRH and stimulate gonadotropin release by the pituitary. CHH may be due to the failure of neuronal migration, defective synthesis or action of GnRH itself, or defects in gonadotroph receptors or secretion. Over 60 individual genes have been identified that disrupt this pathway, including *GNRH1* and *AMHR2* [1]. Some of these are also associated with nonreproductive phenotypes that help to identify the genetic variant involved. The reproductive phenotype in males with CHH may present at birth with features of impaired androgen effect (cryptorchidism and micropenis), in teenage years with the failure of progression into puberty, or still later with infertility and azoospermia. Importantly, GnRH secretion and action are also responsible for minipuberty, which begins soon after birth and continues up to 6 months of age [24]. The foetal GnRH is inhibited by circulating high placental steroids, particularly oestrogen. After the immediate postnatal fall of placental hormones, the infant’s HPG axis is disinhibited, leading to a surge of LH, FSH, and testosterone production in the male infant. Testosterone reaches a peak at 4 to 10 weeks and declines to low or undetectable levels by 6 months. Therefore, infants with CHH are expected to lack the hormonal events of minipuberty, and 4 to 10 weeks after birth is an opportune time to study the functioning of the HPG axis. Moreover, the detection of those hormones, or lack thereof, may provide clues to the genetic milieu or to the probable long-term phenotype of male infants.

*GNRH1* encodes the GnRH decapeptide preprohormone. Case reports of mutations in *GNRH1* in CHH are rare. There have been reported 13 biallelic cases in 10 families, all with severe reproductive phenotype, including cryptorchidism in three out of six males, but not with an associated nonreproductive phenotype [9,25,26,27,28,29,30,31]. There are five digenic case reports with *GNRH1* mutations in combination with *FGFR1* (4 cases) or *DAX1* (1 case) [9,32,33]. Three of these had severe reproductive phenotypes, and in two, the phenotype was unreported. There have been nine monoallelic *GNRH1* mutations reported, with four having a severe reproductive phenotype, including the p.R31C mutation [9,10,11,27,34].

In all, there are only seven cases with CHH reported with the p.R31C mutation, two with an additional variant identified, and the remaining were classed as monoallelic, including one in the same family as cases with digenic aetiology [9,10,11]. Varying degrees of the severity of the reproductive phenotype have been observed in those reported as monoallelic [31]. Frameshift or nonsense mutations in *GNRH1* have been found in recessive inheritance reports [11]. These mutations result in failure to transcribe the GnRH sequence.

Functional studies of the p.R31C mutation have been performed and described in detail by Maione [11]. This mutation produces a single amino acid change at the eighth amino acid in the GnRH decapeptide. The variant affects a highly conserved arginine residue in the GnRH decapeptide sequence, and multiple in silico algorithms support pathogenicity. Amino acid substitutions of histidine, glutamine, leucine, serine, tyrosine, or tryptophan at this position have been shown to reduce the ability of GnRH to bind and activate GnRH receptors [35,36]. It is proposed that the Asn7.45 residue in the GnRH receptor has evolved to specifically recognise the arginine residue in GnRH [37]. Maione et al. performed a comprehensive functional characterisation of this mutant R31C GnRH [11]. Their studies showed that, compared with the wild type, the mutant demonstrated 100-fold lower binding affinity to the receptor; very low activity with SRE luciferase, IP, Ca^2+^, ERK1/2 signalling, and low *Lhb* gene expression and LH secretion. Although on the basis of family studies of three generations of individuals with CHH, the p.R31C mutation has been postulated to be dominant, albeit with a less severe reproductive phenotype than that seen with autosomal recessive frameshift mutations [9], the studies by Maione did not demonstrate negative dominance of wild-type GnRH in vitro [11]. Although sequencing was performed excluding the main genes associated with CHH, the authors conclude that the contribution of another unidentified gene in that study cannot be ruled out.

After detecting minipuberty in the infant, suggesting that the p.R31C *GNRH1* variant he possessed had not impaired his HPG axis, we proceeded to conduct further genetic testing of both father and son using the methods detailed above. This revealed that the father also had a mutation in *AMHR2* p.G445_L453del, a 27 nucleotide deletion of the coding sequence, resulting in a loss of nine amino acids, reported in the literature as “pathogenic”. The *AMHR2* gene variant, also known as rs764761319, is predicted to be potentially disease-causing as the deletion lies within the catalytic intracellular serine/threonine domain of the receptor. This deletion is reported in the Leiden Open Variation Database (LOVD) as pathogenic and is also reported in ClinVar as pathogenic (based on three submissions). Malone et al. concluded that the heterozygous inframe deletion in AMHR2 is a functional change making a potential contribution to the pathogenesis of CHH [38]. However, the authors did not provide a detailed explanation as to how they arrived at the classification of the variant. The application of the ACMG guidelines typically requires access to a variety of information on the variant [13]. Our assessment of the variant using these guidelines yielded the classification of pathogenic. Therefore, our characterisation supports that of Malone [38].

Biallelic *AMHR2* mutations in males are associated with autosomal recessive persistent Müllerian duct syndrome, characterised by normal external male genitalia, variable testicular descent, and the presence of Müllerian structures that may be discovered only incidentally. However, the anti-Müllerian hormone (AMH) and its receptor also have a role in the development and function of GnRH neurons, and defective AMH signalling contributes to the pathogenesis of CHH in humans [38]. AMHR2 is expressed on GnRH neurons in early embryonic development and adulthood in both murine and human brains. AMH is a potent stimulator of GnRH cell motility and the pharmacological or genetic invalidation of AMHR2 signalling in vivo alters GnRH migration and the projections of neurons to the basal forebrain, resulting in reduced size of this neuronal population in adult brains. AMH also stimulates GnRH neuronal activity and hormone secretion in mature GnRH cells [39]. In functional studies, it has been shown that exogenous AMH increases neuronal activity and GnRH secretion and, thereby, gonadotrophin secretion via AMHR2 expressed on GnRH neurons. In a study of 180 probands with CHH, the same mutation in *AMHR2* that we identified was reported by Malone et al. as a sporadic heterozygous mutation in a female with absent puberty and a diagnosis of CHH.

Similarly, functional studies were performed with the *AMHR2* p.Gly445_Leu453del variant and described in detail by Malone et al. [38]. Migration studies were performed using GN11 cells transfected with a plasmid encoding the *hAMHR2*WT or the *AMHR2* p.Gly445_Leu453del mutant, showing migration was significantly impaired with the mutant *AMHR2*. The effect of AMH on GnRH release was studied in GT1-7 cells expressing either *AMHR2*WT or the p.Gly445_Leu453del mutant showing that the mutant *AMHR2* significantly reduced AMH-dependent GnRH secretion compared with the wild type [38].

Therefore, we postulate that, together, the patient’s variants in *AMHR2* and *GNRH1* contributed to CHH by the impaired migration of GnRH neurons, the lack of the AMH effect on GnRH secretion, and altered GnRH decapeptide with reduced binding to GnRH receptors, thereby facilitating a conclusion of digenic disease in this patient. Patil et al. concluded that monoallelic variants in *GNRH1* are questionable in the cause of a clinical phenotype for CHH [31]. Our data support that finding by suggesting that for its manifestation, this monoallelic *GNRH1* p.R31C required the additional deleterious effect from the *AMHR2* functional variant.

In this case, the finding of minipuberty in the infant son, despite having the same *GNRH1* mutation as his father, was instrumental in finding the additional pathogenic *AMHR2* mutation contributing to CHH. We believe that the identification of minipuberty, or lack thereof, via hormonal testing in male infants in the appropriate postnatal window has value for several reasons. First, in male infants with the so-called “red flags” for CHH (cryptorchidism and micropenis), the absence of the hormonal markers of minipuberty may assist in making an early diagnosis of that condition [24]. Secondly, in infants born via assisted reproduction from a parent with CHH, the presence or absence of minipuberty, together with early genetic testing, may be either reassuring or indicate that the child is likely to have a CHH phenotype and prompt timely intervention to induce puberty at the appropriate time. Lastly, if the absence of minipuberty is confirmed, intervention with GnRH or gonadotrophins may induce Sertoli cell proliferation and facilitate fertility as an adult.

This report has limitations. We did not confirm the actions of these two genetic variants together in vitro in terms of GnRH signalling and cell migration using in vitro laboratory methods. However, as mentioned, other authors have performed detailed functional studies on both separately, showing their separate effects on GnRH migration, secretion, and receptor activation [11,38]. Therefore, we consider that the hypothesis that both mutations contribute to the clinical phenotype is well supported by the available data from functional studies. We do not yet know whether the infant will proceed to normal puberty. This report has the strength that we performed exome sequencing of the proband and used complementary approaches to variant classification so as to be sure that no additional aetiological variants would be overlooked. In addition, we took advantage of the appropriate time to test for minipuberty, which led to the identification of the second mutation in the index case and validated our suspicion of digenic disease.

## 6. Conclusions

In this case report, the p.R31C mutation in *GNRH1*, inherited from a father with CHH, was associated with normal minipuberty in the infant son. This led us to the conclusion that the *GNRH1* p.R31C mutation in the heterozygous state did not perturb the HPG axis sufficiently to cause CHH in the patient’s son and did not prevent minipuberty despite the previous clinical advice that this mutation was dominant. The subsequent search and identification of the *AMHR2* p.Gly445_Leu453del variant reinforced our view that the *GNRH1* p.R31C variant is sometimes not sufficient to cause disease and therefore is not certain to be dominant or, at least, exhibits incomplete penetrance and variable expressivity. This report also emphasises the very important opportunity afforded by the time window of minipuberty in assessing inherited genetic disorders of hypothalamic function.

## Figures and Tables

**Figure 1 genes-14-01204-f001:**
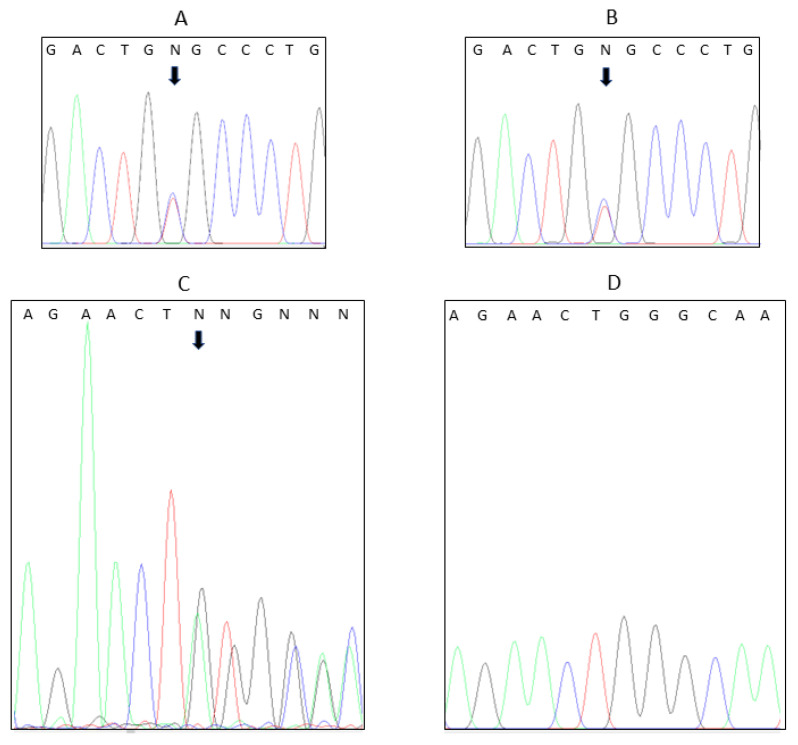
Confirmatory Sanger sequencing chromatograms showing the heterozygous presence of the *GNRH1* missense variant (NM_001083111: c.C91T:p.Arg31Cys) in the patient (**A**) and his son (**B**) and the start of the *AMHR2* 27 nucleotide deletion variant (NM_020547: c.1330_1356del: p.G445_L453del), also present heterozygously in the patient (**C**) but not in his son (**D**) who exhibits normal sequence. Nucleotide peaks, adenine (green), cytosine (blue), guanine (black), thymine (red), are labelled above the chromatogram; N = not able to be determined). The *GNRH1* C91T variants and the beginning of the *AMHR2* 27 nucleotide deletion variant are highlighted by arrows.

**Figure 2 genes-14-01204-f002:**
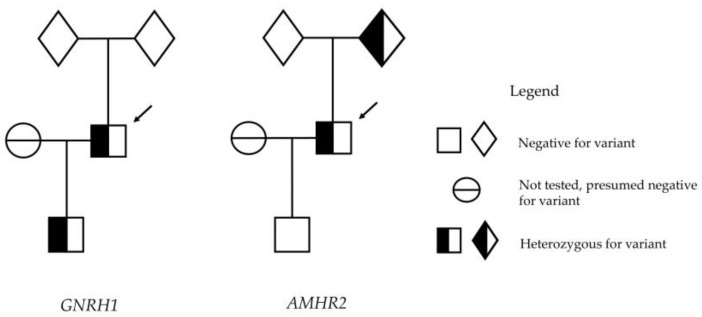
Family trees demonstrating segregation of the *GNRH1* (NM_001083111:exon2: c.C91T:p.Arg31Cys) and *AMHR2* (NM_020547:exon10:c.1330_1356del: p.G445_L453del) variants. The arrows refer to the patient under consideration (index case). Note that the gender of the parents of the index case cannot be disclosed, as indicated by diamond shapes. All patient samples were examined at least twice in separate PCR/sequencing reactions.

## Data Availability

The data presented in this study are available on request from the corresponding author. The data are not publicly available due to patient privacy.

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
