# Peer review of "Digenic Congenital Hypogonadotropic Hypogonadism Due to Heterozygous GNRH1 p.R31C and AMHR2 p.G445_L453del Variants"

_genes, 2023, doi:10.3390/genes14061204_

Round 1
Reviewer 1 Report (New Reviewer)
This CHH case report study by Stuckey et al. is interesting. However, there are some minor comments to improve its quality.
1. The title should be rewritten as in this manuscript the theme is the digenic CHH of the father. Both genes should be present in the title.
2. All parenthesis from mutations should be removed in the abstract and throughout the paper.
3. Too long introduction. Please shorten it.
4. Page 3 line 102. “..other pituitary hormones were normal”, define the other hormones.
5. In case report section, there are some normal values in parenthesis. Determine that they are normal values and not references for example.
6. In methodology, it is not clear how the ΑMHR2 variant was found in the father as he was tested using targeted NGS in 19 other genes.
7. I would recommend adding a pedigree or an image that would explain the methodology and results of this family.
Minor editing required
Author Response
Please see below our response to the reviewer’s suggestions
Reviewer 1 Suggestions for Authors
This CHH case report study by Stuckey et al. is interesting. However, there are some minor comments to improve its quality.
- The title should be rewritten as in this manuscript the theme is the digenic CHH of the father. Both genes should be present in the title.
- We have changed to title to ‘Digenic congenital hypogonadotrophic hypogonadism due to heterozygous GNRH1 p.R31C and AMHR2 p.G445_L453del variants’
- All parenthesis from mutations should be removed in the abstract and throughout the paper.
- Parentheses have been removed
- Too long introduction. Please shorten it.
- We have substantially shortened the introduction
- Page 3 line 102. “..other pituitary hormones were normal”, define the other hormones.
- This has been clarified. Page 3 paragraph 1
- In case report section, there are some normal values in parenthesis. Determine that they are normal values and not references for example.
- We have amended to read in the style LH 0.8 (reference range: 1-5 mU/L) instead of LH 0.8 mU/L (1-5), etc.
- In methodology, it is not clear how the ΑMHR2 variant was found in the father as he was tested using targeted NGS in 19 other genes.
- It seems that the uncertainty arises because it was not made clear that the genetic analyses were done by two separate laboratories. The first analysis for the proband was done by the hospital’s routine laboratory using the panel of genes listed. The second analysis for the proband and that of his son and his parents was done by the specialist molecular genetics laboratory in the Endocrinology Department. We have now made this clear.
- I would recommend adding a pedigree or an image that would explain the methodology and results of this family.
- A pedigree chart has been added
Comments on the Quality of English Language Minor editing required
- English grammar and expression has been reviewed
Reviewer 2 Report (New Reviewer)
This manuscript is a case study for a 28-year-old male and his son, conceived by IVF. The 28-year-old male had previously been diagnosed with a known variant in GNRH1 (pR31C). Interestingly, the male infant inherited this same variant in GNRH1, which had previously been considered a dominant variant, but went through minipuberty with normal testosterone levels at 64 days of age. The authors performed genomic sequencing of a panel of genes for both the proband and the infant, finding a previously described pathogenic 27-nucleotide deletion (non-frameshift) in the AMHR2 gene, only in the proband. The parents of the proband were sequenced and one was found to have the AMHR2 deletion, but neither had the GNRH1 variant.
· A pedigree with genotypes for the family should be included in the manuscript, along with a discussion of the de novo nature of the proband’s GNRH1 variant (i.e. parents did not have this variant).
· The AMRH2 gene name is given, but not its protein name anti-Müllerian hormone (AMH) receptor type 2, early in the results section. There is however a very nice description of AMH in the discussion, but perhaps providing the protein name earlier in the manuscript would be useful for readers.
· The methodology mentions a panel of genes sequenced for CHH, but AMRH2 is not one of those listed. More information on this panel, and the genes sequenced should be provided.
· There are still editing marks (tracking) present in this copy of the manuscript.
Author Response
Please see below the response to Reviewer 2
Reviewer 2
Comments and Suggestions for Authors
This manuscript is a case study for a 28-year-old male and his son, conceived by IVF. The 28-year-old male had previously been diagnosed with a known variant in GNRH1 (pR31C). Interestingly, the male infant inherited this same variant in GNRH1, which had previously been considered a dominant variant, but went through minipuberty with normal testosterone levels at 64 days of age. The authors performed genomic sequencing of a panel of genes for both the proband and the infant, finding a previously described pathogenic 27-nucleotide deletion (non-frameshift) in the AMHR2 gene, only in the proband. The parents of the proband were sequenced and one was found to have the AMHR2 deletion, but neither had the GNRH1 variant.
A pedigree with genotypes for the family should be included in the manuscript, along with a discussion of the de novo nature of the proband’s GNRH1 variant (i.e. parents did not have this variant).
- A pedigree diagram has been included and a further sentence on the conclusion that this is a de novo mutation added.
The AMRH2 gene name is given, but not its protein name anti-Müllerian hormone (AMH) receptor type 2, early in the results section. There is however a very nice description of AMH in the discussion, but perhaps providing the protein name earlier in the manuscript would be useful for readers.
- Thank you. This has been added. First paragraph of results.
The methodology mentions a panel of genes sequenced for CHH, but AMRH2 is not one of those listed. More information on this panel, and the genes sequenced should be provided.
- It seems that the uncertainty arises because it was not made clear that the genetic analyses were done by two separate laboratories. The first analysis for the proband was done by the hospital’s routine laboratory using the panel of genes listed. The second analysis for the proband and that of his son and his parents was done by the specialist molecular genetics laboratory in the Endocrinology Department. We have now made this clear.
We have listed the panel of genes sequenced by the routine laboratory in the first paragraph of Methods. It did not include AMHR2. The AMHR2 variant was found by the specialist molecular genetics laboratory in the Endocrinology Department by the methods described in the second and third paragraph of Methods.
There are still editing marks (tracking) present in this copy of the manuscript.
- These have been removed and only the new editing tracks remain
This manuscript is a resubmission of an earlier submission. The following is a list of the peer review reports and author responses from that submission.
Round 1
Reviewer 1 Report
The submitted manuscript by Stuckey et al entitled “Minipuberty in an infant inheriting the GNRH1 R31C mutation 2 uncovers digenic congenital hypogonadotropic hypogonadism 3
in the father” that describes a case report has been written according to requirements of such studies. There are only some minor concerns:
1. The authors should present a statement of ethical approval of the study by an Ethical Committee and the specified approval code, if available.
2. Sequence information must be deposited to the appropriate database and accession numbers provided by the database should be included in the submitted manuscript.
3. Confirmation of whole exome sequencing by Sanger sequencing should be presented in a figure. Even if there is a limitation for presentation in the main text of the case report, it should be presented as supplementary data.
Author Response
Reviewer 1
- The authors should present a statement of ethical approval of the study by an Ethical Committee and the specified approval code, if available.
- No ethical approval was required for this study since it was conducted as part of patient care. Informed consent was provided by the subjects and these details are now noted in the paper.
- Sequence information must be deposited to the appropriate database and accession numbers provided by the database should be included in the submitted manuscript.
- The genetic variants reported in this manuscript were previously known and were present in LOVD – that detail has been added to the manuscript as suggested.
- Confirmation of whole exome sequencing by Sanger sequencing should be presented in a figure. Even if there is a limitation for presentation in the main text of the case report, it should be presented as supplementary data
- Figure 1 showing the Sanger sequence traces is included as suggested.

Reviewer 2 Report
In this study Stuckey et al., reported “Minipuberty in an infant inheriting the GNRH1 R31C mutation 2 uncovers digenic congenital hypogonadotropic hypogonadism 3 in the father”. Overall, in regard to genetic aspect the case is poorly presented. I have some major comments for the improvement of the paper.
Major comments:
Point 1. The paper can be written in more clear way.
Point 2. Nomenclature of the variants should be corrected throughout the manuscript. It is not mentioned what kind of sequencing was performed for the identification of variant GNRH1:NM_001083111:exon2:c.C91T:p.Arg31Cys)?
Point 3. Later on, whole exome sequencing (WES) was performed, kindly provide detail of the WES, how filtration was applied? How many variants were identified and how variants were filter out?
Point 4. Provide chromatogram of Sanger sequencing.
Point 5. Provide ClinVar ID or Leiden Open Variation Database ID
Point 6. Both GNRH1 OMIM 614841 and AMHR2 OMIM 261550 are established to be autosomal recessive, so to establish it as digenic heterozygous, the identified variants should be supported by functional/ in vitro studies.
Point 7. As they went through IVF, then why zygotes were not checked for PGD testing?
Author Response
Reviewer 2
Point 1. The paper can be written in more clear way.
- The paper has been extensively revised to make the methods clearer
Point 2. Nomenclature of the variants should be corrected throughout the manuscript. It is not mentioned what kind of sequencing was performed for the identification of variant GNRH1:NM_001083111:exon2:c.C91T:p.Arg31Cys)?
- Human Genome Variation Society nomenclature is now used for the variants – full nomenclature when first stated in the text, thereafter the accepted abbreviation is used. Abbreviation is used in the abstract for brevity. Sequencing methods have been expanded as suggested.
Point 3. Later on, whole exome sequencing (WES) was performed, kindly provide detail of the WES, how filtration was applied? How many variants were identified and how variants were filter out?
- Details have been expanded
Point 4. Provide chromatogram of Sanger sequencing.
- Figure 1 showing the Sanger sequence traces is included as suggested.
Point 5. Provide ClinVar ID or Leiden Open Variation Database ID
- LOVD variant numbers have been included as suggested.
Point 6. Both GNRH1 OMIM 614841 and AMHR2 OMIM 261550 are established to be autosomal recessive, so to establish it as digenic heterozygous, the identified variants should be supported by functional/ in vitro studies.
- Functional studies have been done by authors cited in the manuscript for the separate variants. Specifically, Maione et al for GNRH1and Malone et al for AMHR2. We have acknowledged in the manuscript that we have not yet done functional/in vitro studies of the combination – such work would require a successful application for grant funding and extensive molecular genetics experiments. Therefore, given the existing data provided by the two papers we cited, we consider it is better to make this information public now to aid other clinicians and researchers, rather than delay publication for an extended period, when the basic functional data cited is already available.
Point 7. As they went through IVF, then why zygotes were not checked for PGD testing?
- The local Reproductive Technology Council has criteria for approval of PGD of embryos. CHH would not meet the criteria for severity and disability of disease. A sentence has been added to the manuscript to address this point.
Round 2
Reviewer 2 Report
Over all the paper is improved but still I have some major concern about the data. Specifically I have concern about the analysis of the identified variant, maybe the real causative variant is missed. Some of the examples are below.
Point 1. In the abstract it is mentioned that “the GNRH1 p.(R31C) mutation, reported in the literature as pathogenic and dominant” while few lines latter they mentioned that “This leads us to the conclusion that the observed GNRH1 R31C mutation in the heterozygous state is not certain to be dominant”. In reality this variant is reported as variant of unknown significance (VUS) in LOVD.
Point 2. In the lines 133 – 135 it is stated that variant of GNRH1 is de novo in the patients i.e. 28 years old father but they did not mentioned whether the parents of this 28 years old man were tested for the variant?
Point 3. The following statement contradict to the statement in “response to reviewer.” In the lines 136 – 138 they stated that “We postulate, therefore, on the basis of the genetics and the clinical phenotype, that CHH in the patient is digenic and on the basis of the post-natal biochemistry of his son, that the GNRH1 p.(R31C) gene mutation is not dominant”
Point 4. Although variants annotation and bioinformatic analysis are performed very carefully but variant filtration criteria is not satisfactory from the stage of 89,027 variants, maybe the real causative genetic variant is missed during the analysis.
Point 5. Patil et al., (cited as reference 4 in this manuscript) also concluded that monoallelic variant in GNRH1 are questionable to cause the phenotype, their exact statement is “GNRH1 biallelic variants lead to severe reproductive phenotype, with low gonadotropin levels without nonreproductive features or oligogenicity. However, the role of GNRH1 monoallelic variants in CHH pathophysiology for reported variants remains questionable.”
Point 6. Malone et al., (cited as reference 8 in this manuscript) concluded that “The conclusion from the paper is that the 3 variants identified in AMH and the single variant found in AMHR2 are pathogenic. However, the authors do not provide adequate explanation as to how they arrived at the classification of these variants… under ACMG guidelines, all of the variants reported in this paper would be classified as variants of uncertain significance”. According to my assessment the reported variant of AMHR2 should be classified as variant of unknown significance.
Point 7. Reported variant of AMHR2 is more than 100 times reported in public databases.
Point 8. I would suggest reanalyzing the data of whole exome sequencing and put a supplementary table in the manuscript.
Author Response
Thank you for these suggestions and helpful comments. Responses are provided in context below.
Apart from clarifying the validity of the genetic analysis we would like to highlight the very important point we are making. That is that the identification of minipuberty, or lack thereof, by hormonal testing in male infants in the appropriate post-natal window, allows an early-in-life estimation of the functioning of the hypothalamic-pituitary-gonadal axis in cases of known genetic variants in the parent.
Reviewer points:
Over all the paper is improved but still I have some major concern about the data. Specifically I have concern about the analysis of the identified variant, maybe the real causative variant is missed. Some of the examples are below.
- Following this suggestion, we have completed reanalysis of the exome data, both filtered and unfiltered, manually and with Phenolyzer software (Yang, et al. Nature Methods, 2015 12:841-843), and confirm that nothing has been missed. A statement to this effect has been added to the text.
The GNRH1 variant was the only result that achieved an integrated score >0.5 for the CHH phenotype.
The AMHR2 variant cannot be modelled by Phenolyzer because there is no RadialSVM score or MetaSVM rankscore score available.
Point 1. In the abstract it is mentioned that “the GNRH1 p.(R31C) mutation, reported in the literature as pathogenic and dominant” while few lines latter they mentioned that “This leads us to the conclusion that the observed GNRH1 R31C mutation in the heterozygous state is not certain to be dominant”. In reality this variant is reported as variant of unknown significance (VUS) in LOVD.
- The GNRH1(R31C) variant is published as pathogenic (Maione L, et al. PLoS One. 2013 Jul 25;8(7):e69616) and reported in Varsome as pathogenic by ACMG.
Furthermore, we also independently classified the variant as pathogenic, using the Genetic Variant Interpretation Tool (University of Maryland), which efficiently implements the ACMG Guidelines.
This detail is in the text.
Therefore in the manuscript, we stated that GNRH1 p.(R31C) is reported in the literature as pathogenic, including functional proof. This had been conveyed to the patient by the geneticist and led to the testing of the infant at birth. However, the observed effects in our patient and son highlight that there is a reason to question the prior conclusion of dominance. We acknowledge that different resources often report different levels of pathogenicity interpretation for variants.
Point 2. In the lines 133 – 135 it is stated that variant of GNRH1 is de novo in the patients i.e. 28 years old father but they did not mentioned whether the parents of this 28 years old man were tested for the variant?
- We did state that the variant is de novo, both parents are wildtype for the GNRH1(R31) and paternity and maternity were confirmed. The variant has unquestionably arisen de novo in this patient. The text does state this as follows:- Family studies confirmed the AMHR2 p.(G445_L453del) variant in one of the patient’s parents, but neither had the GNRH1 p.(R31C) variant.
Point 3. The following statement contradict to the statement in “response to reviewer.” In the lines 136 – 138 they stated that “We postulate, therefore, on the basis of the genetics and the clinical phenotype, that CHH in the patient is digenic and on the basis of the post-natal biochemistry of his son, that the GNRH1 p.(R31C) gene mutation is not dominant”
- We have reworded this paragraph to make the meaning clearer as follows:-
We postulate, therefore, on the basis of his genetics and clinical phenotype, that CHH in the patient is digenic, produced by the combination of the GNRH1 and AMHR2 variants. Secondly, on the basis that his son inherited the GNRH1 p.(R31C) variant only and yet achieved minipuberty, we postulate that this variant is pathogenic only under certain circumstances, and therefore appears to not be dominant.
Point 4. Although variants annotation and bioinformatic analysis are performed very carefully but variant filtration criteria is not satisfactory from the stage of 89,027 variants, maybe the real causative genetic variant is missed during the analysis.
- We have added further detail to clarify this point – “ Reanalysis of the exome data, both filtered and unfiltered, manually and with Phenolyzer software {Yang, 2015 #4955} was performed to confirm that no other relevant pathogenic variants were overlooked. Manual curation involved sorting and filtering based on minor allele frequency and metrics available from ANNOVAR analysis, including Polyphen, SIFTS, MutationAssessor, CADD, GERP, SiPhy. Phenolyzer analysis employs a machine learning model that integrates multiple features to score and prioritize all genes potentially relevant to the patient’s disorder, in combination with variant pathogenicity meta predictors. It therefore seems very unlikely that any other relevant pathogenic variants have been overlooked.”
Point 5. Patil et al., (cited as reference 4 in this manuscript) also concluded that monoallelic variant in GNRH1 are questionable to cause the phenotype, their exact statement is “GNRH1 biallelic variants lead to severe reproductive phenotype, with low gonadotropin levels without nonreproductive features or oligogenicity. However, the role of GNRH1 monoallelic variants in CHH pathophysiology for reported variants remains questionable.”
- We have modified the text to acknowledge this point.
Patil et al., concluded that monoallelic variants in GNRH1 are questionable in the cause of a clinical phenotype for CHH, and our data would appear to support that, since despite receipt of a diagnostic genetics report that the GNRH1 p.(R31C) was pathogenic, the clinical and familial data we have available suggests that to be manifest, this monoallelic GNRH1 p.(R31C) required the additional digenic effect from the AMHR2 functional variant.
Point 6. Malone et al., (cited as reference 8 in this manuscript) concluded that “The conclusion from the paper is that the 3 variants identified in AMH and the single variant found in AMHR2 are pathogenic. However, the authors do not provide adequate explanation as to how they arrived at the classification of these variants… under ACMG guidelines, all of the variants reported in this paper would be classified as variants of uncertain significance”. According to my assessment the reported variant of AMHR2 should be classified as variant of unknown significance.
- We have modified the text to address this point.
Malone et al., (8) concluded that the heterozygous inframe deletion in AMHR2 is a functional change making a potential contribution to the pathogenesis of CHH. However, the authors do not provide a detailed explanation as to how they arrived at the classification of the variant. Application of the ACMG guidelines typically requires access to a variety of information on the variant. Our assessment of the variant using these guidelines yielded the classification of pathogenic. Therefore our characterisation supports that of Malone.
Furthermore, we note that there are 3 other reports of this variant in ClinVar (VCV000008627.3) all of which classify it as pathogenic (criteria provided, multiple submitters and no conflicts).
Point 7. Reported variant of AMHR2 is more than 100 times reported in public databases.
- We note that the minor allele frequency (MAF) reported by GnomAD (MAF = 0.00040), EXAC (0.000478), UK10K (0.000397) among others, for this variant, show it is rare. We acknowledge 1 unusual report of a small sample from Vasterbotten in Northern Sweden, SweGen (n=300 individuals), reporting the variant at slightly higher frequency. This appears to be an outlier/anomaly compared to the rest of the available data. Note that, as mentioned, this AMHR2 variant is reported as pathogenic in ClinVar.
Point 8. I would suggest reanalyzing the data of whole exome sequencing and put a supplementary table in the manuscript.
- Thank you for this suggestion. We did this and found no additional compelling variants of interest. The patient consent does not permit us to publish data other than variants that are relevant to the central theme of the paper and were reported by clinical diagnostic laboratories.
We thank the reviewers for their assistance with this manuscript and hope that the editorial team will now consider it suitable for publication in Genes.

Round 3
Reviewer 2 Report
As I mentioned in detail in my last review, that "In conclusion in the current form, it will be difficult to accept the two variants responsible for the phenotype, however supportive functional studies can make the story acceptable. Publishing the manuscript in current form can be misleading."